# Independent regulation of age associated fat accumulation and longevity

Anthony O. Beas [1✉], Patricia B. Gordon[1,3], Clara L. Prentiss[1], Carissa Perez Olsen[1,4], Matthew A. Kukurugya[2,5], Bryson D. Bennett[2], Susan M. Parkhurst [1] & Daniel E. Gottschling [2]

Age-dependent changes in metabolism can manifest as cellular lipid accumulation, but how this accumulation is regulated or impacts longevity is poorly understood. We find that *Saccharomyces cerevisiae* accumulate lipid droplets (LDs) during aging. We also find that over-expressing *BNA2*, the first Biosynthesis of NAD$^+$ (kynurenine) pathway gene, reduces LD accumulation during aging and extends lifespan. Mechanistically, this LD accumulation during aging is not linked to NAD$^+$ levels, but is anti-correlated with metabolites of the <u>s</u>hikimate and <u>a</u>romatic amino acid biosynthesis (SA) pathways (upstream of *BNA2*), which produce tryptophan (the Bna2p substrate). We provide evidence that over-expressed *BNA2* skews glycolytic flux from LDs towards the SA-BNA pathways, effectively reducing LDs. Importantly, we find that accumulation of LDs does not shorten lifespan, but does protect aged cells against stress. Our findings reveal how lipid accumulation impacts longevity, and how aging cell metabolism can be rewired to modulate lipid accumulation independently from longevity.

[1] Basic Sciences Division, Fred Hutchinson Cancer Research Center, Seattle, WA 98109, USA. [2] Calico Life Sciences LLC, South San Francisco, CA 94080, USA. [3] Present address: Laboratory of Cell Biology, Center for Cancer Research, National Cancer Institute, National Institutes of Health, Bethesda, MD 20892, USA. [4] Present address: Worcester Polytechnic Institute, Department of Chemistry and Biochemistry, 60 Prescott St, Worcester, MA 01605, USA. [5] Present address: Department of Molecular and Cell Biology, University of California Berkeley, Berkeley, CA 94720, USA. ✉email: abeas@fredhutch.org

As most organisms age, metabolism slows, which can manifest as a gradual accumulation of neutral lipid (i.e., fat), an important energy source stored in lipid droplets (LDs). Aspects of metabolic decline during aging have been attributed to an age-associated decline in nicotinamide adenine dinucleotide ($NAD^+$) levels[1,2]. Intense research has focused on interventions to increase cellular $NAD^+$, such as with $NAD^+$ precursors (e.g., niacin, nicotinamide mononucleotide, and nicotinamide riboside), and to mitigate age-associated changes in metabolism, including reducing fat. While such interventions have been shown to improve facets of metabolism and reduce fat, they do not always increase $NAD^+$ or extend lifespan[3–7]. Recent studies present alternative mechanisms that modulate fat accumulation during aging. For example, it was shown that during aging in mice, the frequency of DNA double stranded breaks increases and this activates a muscle specific enzyme (DNA-PK) that is responsible for 40% of the weight gained by mice fed high-fat diets[8]. In aging flies, specific muscles accumulate LDs, and the overexpression of a cytosolic histone deacetylase (HDAC6) suppresses this accumulation[9]. Thus, many questions remain concerning how lipid accumulation is regulated during aging and specifically, whether this accumulation during aging reduces longevity.

Here, we explore whether LDs change in *Saccharomyces cerevisiae* cells as they replicatively age, and find that LDs accumulate. We describe a new link between the biosynthesis of $NAD^+$ (kynurenine) pathway and lipid droplets during aging. Specifically, our genetic and metabolomic approaches reveal that increasing the BNA pathway (by overexpressing *BNA2*, the first gene of the pathway) reduces LD accumulation during aging. This reduction is achieved by pulling metabolic flux, likely from glycolysis, through the shikimate and aromatic amino acid biosynthesis (SA) pathways, which are upstream of the BNA pathway and which synthesize and supply tryptophan, the major substrate of the BNA pathway. By using genetic approaches to dissect the role of the SA-BNA pathway on longevity, we find, surprisingly, that LD accumulation during aging can be modulated independently from longevity. Thus, LD accumulation during normal aging does not shorten lifespan. Rather, we find LD accumulation protects aging cells against cold stress. These findings reveal how lipid-droplet accumulation impacts longevity, and provide a new strategy for lessening lipid accumulation during aging independently from longevity.

## Results

**Lipid droplets accumulate during yeast replicative aging**. Replicative aging in the budding yeast, *Saccharomyces cerevisiae*, is defined as the number of times an individual cell divides before death[10]. We previously showed that replicatively aged yeast cells exhibit characteristics of age-induced decline observed in metazoa, including reduced mitochondrial function and increased genomic instability[11,12]. In this study we screened cells for age-associated changes in organelles and vesicles[12,13], and found that lipid droplets (LDs) accumulate (Fig. 1a). Quantification of LDs using BODIPY 493/503 revealed LDs increase 7.2-fold from median age 0 (young) to 23 (old) (Fig. 1c, d, Supplementary Fig. 1), which was corroborated biochemically (Fig. 1b).

**A link between the biosynthesis of $NAD^+$ and lipid droplets**. To gain insights into how this LD accumulation might be regulated, we performed a limited over-expression screen for reduced LDs that included genes previously associated with yeast glycerolipid and LD dynamics (see Supplementary Data 1)[14–17]. We identified *BNA2*, which was not previously associated with regulation of yeast LDs, and encodes an indoleamine 2,3-dioxygenase that utilizes tryptophan in the first step of the biosynthesis of $NAD^+$ (or kynurenine) pathway[18]. We made homozygous, *BNA2* overexpression diploid cells (*BNA2*-OE cells), which we compared with diploid WT (control) cells throughout this study. In young *BNA2*-OE and control cells, LD levels were similar, whereas in old *BNA2*-OE cells, LD levels were significantly decreased: a 40% reduction compared with control cells not over-expressing *BNA2* (Fig. 1c, d, Supplementary Fig. 1), indicating *BNA2*-OE reduces LD levels in aged cells. To determine whether *BNA2*-OE impacts replicative lifespan, we used complementary approaches [microdissection[10] and a new protocol applying the Mother Enrichment Program[19] (see Methods)], and found in each case that BNA2-OE cells exhibit significant increases of lifespan (>15%) (Fig. 1e, f, Supplementary Fig. 2a). These data are consistent with the simple hypothesis that reducing LD accumulation during aging promotes longevity.

Given Bna2p's role, it was anticipated that *BNA2*-OE would affect $NAD^+$ levels; however, we observed no significant difference in $NAD^+$ levels between *BNA2*-OE and control cells (Figs. 1g–h, 4f). To determine how the BNA pathway affects LD accumulation and longevity, we made homozygous deletions in core BNA pathway genes that act "downstream" of *BNA2* (*BNA1*, *BNA2*, *BNA5*, *BNA6*, *BNA7*) and a branch point gene (*BNA3*) (Figs. 2a, 3a)[18,20]. Deleting core BNA pathway genes significantly reduced lifespan (Fig. 2h, j–l, Supplementary Figs. 2b,d,e, 3b), whereas deleting *BNA3* had no effect (Fig. 2i, Supplementary Fig. 2c). These data indicated that the core BNA pathway, but not the *BNA3* branch point, was important for longevity. However, even though deleting core BNA pathway genes reduced lifespan (Fig. 2h, j–l, Supplementary Figs. 2b,d,e, 3b), these deletions did not affect the normal accumulation of LDs during aging (Fig. 2b, d–f, Supplementary Fig. 3a), which is inconsistent with the simple hypothesis noted above, but rather suggests that LD accumulation during aging does not impact lifespan.

**Lipid droplet accumulation is separable from longevity**. We investigated whether BNA pathway genes were required for the ability of *BNA2*-OE to reduce LD accumulation during aging. Strikingly, cells with overexpressed *BNA2* still displayed reduced LD accumulation during aging regardless of whether BNA pathway core or branch point genes were deleted (*bna1Δ*, *bna3Δ*, *bna5Δ*, *bna6Δ*, or *bna7Δ*) (Fig. 2b–f). These results suggested that the flux of metabolites from Bna2p toward $NAD^+$ synthesis may not be important for suppressing the age-associated accumulation of LDs by *BNA2*-OE.

We next investigated the dependency of the BNA pathway on longevity in *BNA2*-OE cells. Cells with *BNA2*-OE combined with either *bna1Δ*, *bna3Δ*, or *bna5Δ* deletions had significantly increased lifespan compared with the cells with only the respective BNA pathway genes deleted, i.e., *bna1Δ*, *bna3Δ*, or *bna5Δ* (Fig. 2h–j, Supplementary Fig. 2b–d). Interestingly, the increase in lifespan of *BNA2*-OE cells was to the same degree (above WT (control) cells) with or without *BNA3* (Fig. 2i), but the increase in lifespan from *BNA2*-OE only achieved control levels when either *BNA1* or *BNA5* were eliminated (Fig. 2h, j, Supplementary Fig. 2b,d). In further contrast, deleting *BNA6* or *BNA7* blocked the ability of *BNA2*-OE to extend lifespan beyond the shortened lifespan of *bna6Δ* and *bna7Δ* cells, respectively (Fig. 2k–l, Supplementary Fig. 2e). Thus, *BNA2*-OE can extend lifespan without *BNA1*, *BNA3*, or *BNA5* but not without *BNA6* or *BNA7*. These data indicate that the BNA pathway is linked to lifespan extension, but is not necessary for suppressing LD accumulation during aging, by *BNA2*-OE. Taken together, our results indicate that LD accumulation and longevity are genetically separable and can be modulated independently.

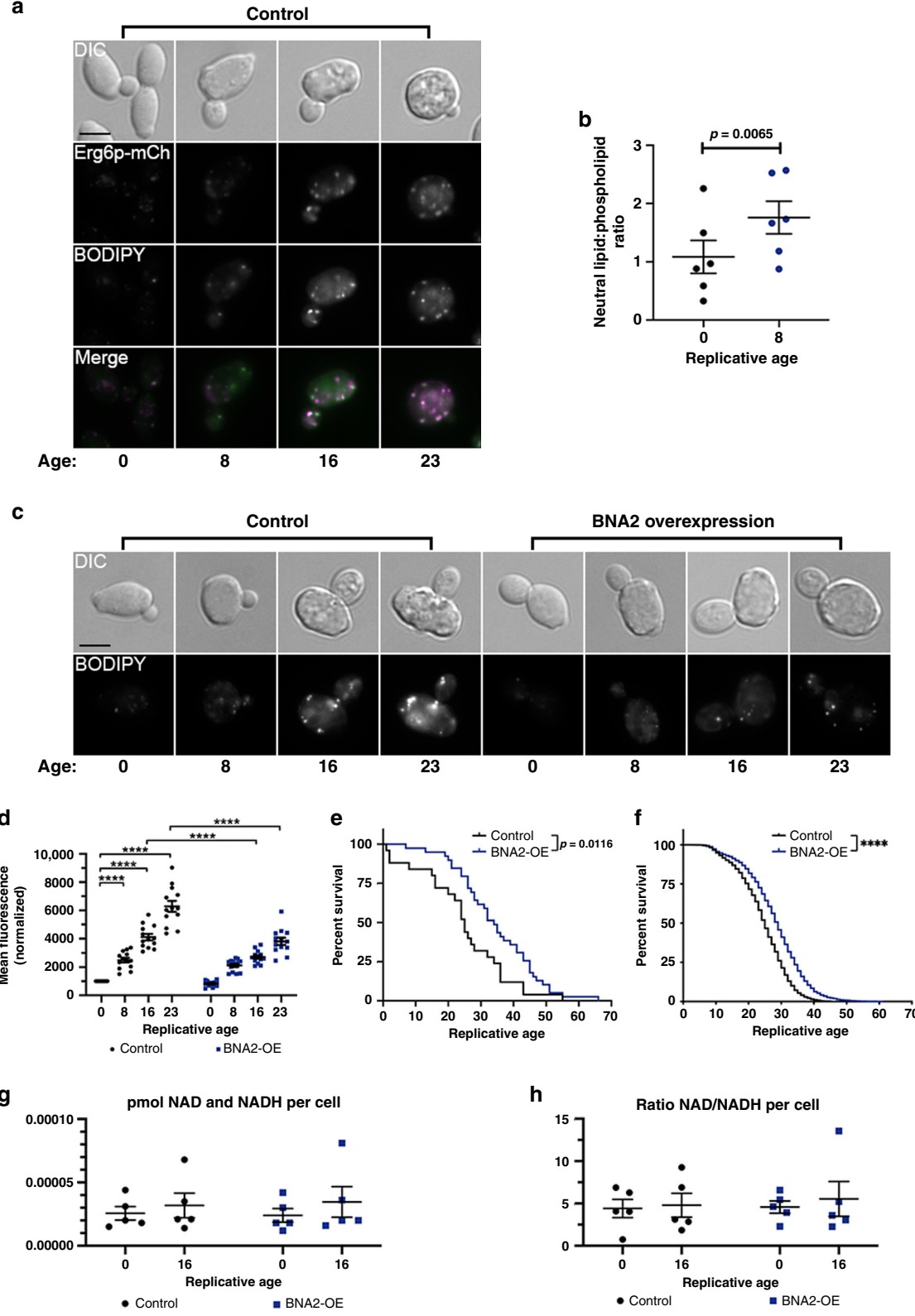

Since BNA pathway genes downstream of *BNA2* were not required for the suppression of age-associated LD accumulation by *BNA2*-OE, we examined whether the shikimate and aromatic amino acid synthesis (SA) pathways upstream of *BNA2* were critical for this phenotype. Substrates for the shikimate pathway are supplied by glycolysis (phosphoenolpyruvate) and the pentose phosphate pathway (erythrose-4-phosphate)[21,22].

These substrates are converted into chorismate, the precursor of the aromatic amino acids phenylalanine, tyrosine, and the Bna2p substrate, tryptophan (Figs. 2a, 3a). There is also crosstalk between the aromatic amino acid synthesis and BNA pathways, through a shared intermediate, anthranilate[23], and a gene, *ARO9*[24] (Fig. 3a, purple). *ARO1* is a gene upstream of aromatic amino acid synthesis that encodes the rate limiting enzyme of the

**Fig. 1 Age-associated lipid droplet accumulation is suppressed by *BNA2* overexpression. a**, **c** Replicative age below panels represents the median age of mother cells determined by budscar counting (see Methods). **a** Lipid droplets (LDs) in Control (WT, AB18-07) cells visualized by mCherry tagged Erg6p, a known yeast LD protein (magenta) and BODIPY 493/503 (neutral lipid stain, green) (black scale bar: 4 μm). Representative images from four independent experiments. **b** Total neutral lipid and phospholipid fractions extracted from young (black dots) and aged (blue dots) UCC4925[12, 13, 19] cells were analyzed by gas chromatography mass spectrometry ($n = 6$ independent experiments). Paired $t$-test (two-tailed): **$p = 0.0065$. **c** BODIPY stained LDs in Control (WT, AB18-07) and *BNA2* overexpression (*BNA2*-OE) cells (black scale bar: 4 μm). Representative images were taken from cells aged and quantified in one experiment from (**d**) below. **d** LDs were stained with BODIPY 493/503 and quantified in Control (WT, AB18-07, black dots) and *BNA2*-OE (blue squares) cells by flow cytometry. The mean, normalized BODIPY intensities for each time point and strain were determined as in Supplementary Fig. 1 for each individual experiment, and these mean, normalized intensities were averaged for $n = 13$ independent experiments and plotted ±SEM. Two-way ANOVA multiple comparisons test: ****$p \leq 0.0001$. $p > 0.05$ indicated as ns (not significant). **e** Replicative lifespan (RLS) determined using microdissection[10]. 25 Control cells (WT, AB18-07, black line, median lifespan 25, maximum lifespan 55) and 39 *BNA2*-OE cells (blue line, median 34, maximum 66) analyzed in $n = 1$ independent experiment. Log-rank test: $p = 0.0116$. **f** RLS determined using the Mother Enrichment Program[19] (see Methods). 2300 Control cells (WT, AB18-07, black line) and 2450 *BNA2*-OE cells (blue line) analyzed over $n = 16$ independent experiments. Average median and maximum lifespan (±SEM): Control (25.2 ± 0.3, 41.6 ± 0.6), *BNA2*-OE (28.9 ± 0.5, 49.9 ± 1.3). Log-rank test: ****$p \leq 0.0001$. **g**, **h** Cellular NAD$^+$ and NADH extracted from Control (WT, AB18-07, black dots) and *BNA2*-OE (blue squares) cells were quantified by absorbance at 450 nm, averaged, and plotted ±SEM ($n = 5$ independent experiments). Source data for (**b**), (**d**–**h**) provided as a Source data file.

shikimate pathway (Fig. 3a). When *ARO1* was deleted, this blocked the ability of *BNA2*-OE to suppress LD accumulation during aging (Fig. 2g), indicating that *ARO1* is essential for *BNA2*-OE to suppress LDs in aging cells. With respect to lifespan, deleting *ARO1* was more nuanced. In the absence of *ARO1*, cells had a dramatically reduced median lifespan compared with WT (control) cells (Fig. 2m). Yet *BNA2*-OE increased this shortened lifespan of *aro1Δ* cells to longer than WT lifespan levels, albeit not as long as *BNA2*-OE alone (Fig. 2m, Supplementary Fig. 2f). Taken together, these results suggested that *BNA2*-OE reduced LD accumulation during aging by pulling upstream substrates away from LDs through the upstream SA pathway, and extended lifespan by increasing flux downstream toward the BNA pathway (Fig. 3j).

**Metabolic rewiring reduces lipid droplet levels in old cells.** To test this idea about how *BNA2*-OE impacts LD accumulation, metabolites within the SA-BNA pathway (Fig. 3a) were examined in some of the aforementioned genetically altered cells (see Supplementary Figure 4 and Source Data file). Although *BNA2*-OE minimally affected phosphenolpyruvate levels (Fig. 3b), *BNA2*-OE dramatically increased SA pathway metabolite levels, including shikimate, shikimate-3-phosphate, and chorismate (by ~16 to ~60-fold) (Fig. 3b, c), and to a lesser extent tyrosine, phenylalanine, and tryptophan levels (by ~4-fold, ~1.9-fold, and ~47%, respectively) (Fig. 3b, e). Chorismate and anthranilate were also increased markedly in *BNA2*-OE *bna6Δ* cells (Fig. 3c,d), which have reduced levels of LDs (Fig. 2e); however, these metabolites were at low or background levels in *BNA2*-OE *aro1Δ* cells (Fig. 3c,d), which have high levels of LDs (Fig. 2g). Thus, *BNA2*-OE requires *ARO1* to increase SA pathway metabolites, and importantly, SA pathway metabolite levels inversely correlate with LD accumulation during aging.

In the early steps of the core BNA pathway, *BNA2*-OE greatly increased kynurenine and 3-hydroxykynurenine (3-HK) levels (by >170-fold) (Fig. 3b, f, h). These metabolites were also detected at high levels in *BNA2*-OE *bna6Δ* and *BNA2*-OE *aro1Δ* cells (Fig. 3f, h), though there was less kynurenine (~20-fold less) and 3-HK (~2.7-fold less) in *BNA2*-OE *aro1Δ* than *BNA2*-OE cells (Fig. 3f, h), consistent with kynurenine and 3-HK being derived from exogenous tryptophan in the media. Importantly, these results are consistent with *BNA2*-OE increasing core BNA pathway metabolite levels (through 3-HK) by pulling substrates through the upstream SA pathway. Notably, *BNA2*-OE minimally altered metabolite levels at the last steps of the core BNA pathway [3-hydroxyanthranilate (3-HA, increased ~2.9-fold) (Fig. 3b) and quinolinic acid (minimal change) (Fig. 3b)] and NAD$^+$ (minimal change) (Figs. 1e, f, 3i). Thus, *BNA2*-OE greatly increases core BNA pathway metabolite levels through the 3-HK step of the pathway, but has much less effect on the steps that proceed thereafter in the production of NAD$^+$. Thus, our results suggest that through 3-HK, the metabolic flux can be diverted into BNA branch points and not necessarily toward NAD$^+$.

Consistent with this idea, *BNA2*-OE induced changes also increased branch point metabolites associated with kynurenine and 3-HK (Fig. 3a). Specifically, *BNA2*-OE increased kynurenic acid and xanthurenate ~350- and 49-fold, respectively (Fig. 3b). Notably, kynurenic acid was also high in *BNA2*-OE *bna6Δ* cells (Fig. 3g), but not in *BNA2*-OE *aro1Δ* cells (Fig. 3g). This latter finding is consistent with increased branch point metabolite levels requiring increased levels of tryptophan, formylkynurenine, or kynurenine and/or crosstalk between the aromatic amino acid synthesis and BNA pathways (Fig. 3a, purple)[23,24]. Taken together, these data strongly support the model that *BNA2*-OE reduces LD accumulation during aging by pulling substrates through the SA pathway toward the BNA pathway and away from LDs (Fig. 3j).

**Lipid droplets may protect aging cells against stress.** LDs have been implicated in offering protection from several types of cellular stresses[25]. This led us to explore whether the accumulation of LDs in aging cells provides any benefit to old cells. This idea was tested with a cold-shock regimen: cell viability was determined in middle-aged cells after exposure to cold (4 ℃). Interestingly, cold exposure slightly increased the lifespan of WT (control) cells (Fig. 4a, Supplementary Fig. 3g), and minimally affected *BNA2*-OE *aro1Δ* cells (Fig. 4b, Supplementary Fig. 3h). However, cold exposure significantly decreased the lifespan of *BNA2*-OE cells (Fig. 4c, Supplementary Fig. 3i). Because aged control and *BNA2*-OE *aro1Δ* cells accumulate more LDs, and aged *BNA2*-OE cells accumulate fewer LDs (Fig. 2g), we speculate that LD accumulation during aging offers a level of protection against stress.

## Discussion

High-fat diets contribute to fat accumulation and disease (e.g., obesity, cancer)[26], but how fat accumulated during normal aging impacts health and longevity is less understood. The present study shows that yeast normally accumulate fat in the form of LDs during aging, similar to metazoans[8,9]. We find that LD accumulation during aging does not simply correlate with longevity (Fig. 2), but does associate with longevity under stress (Fig. 4). Similar context specific correlations between LDs and

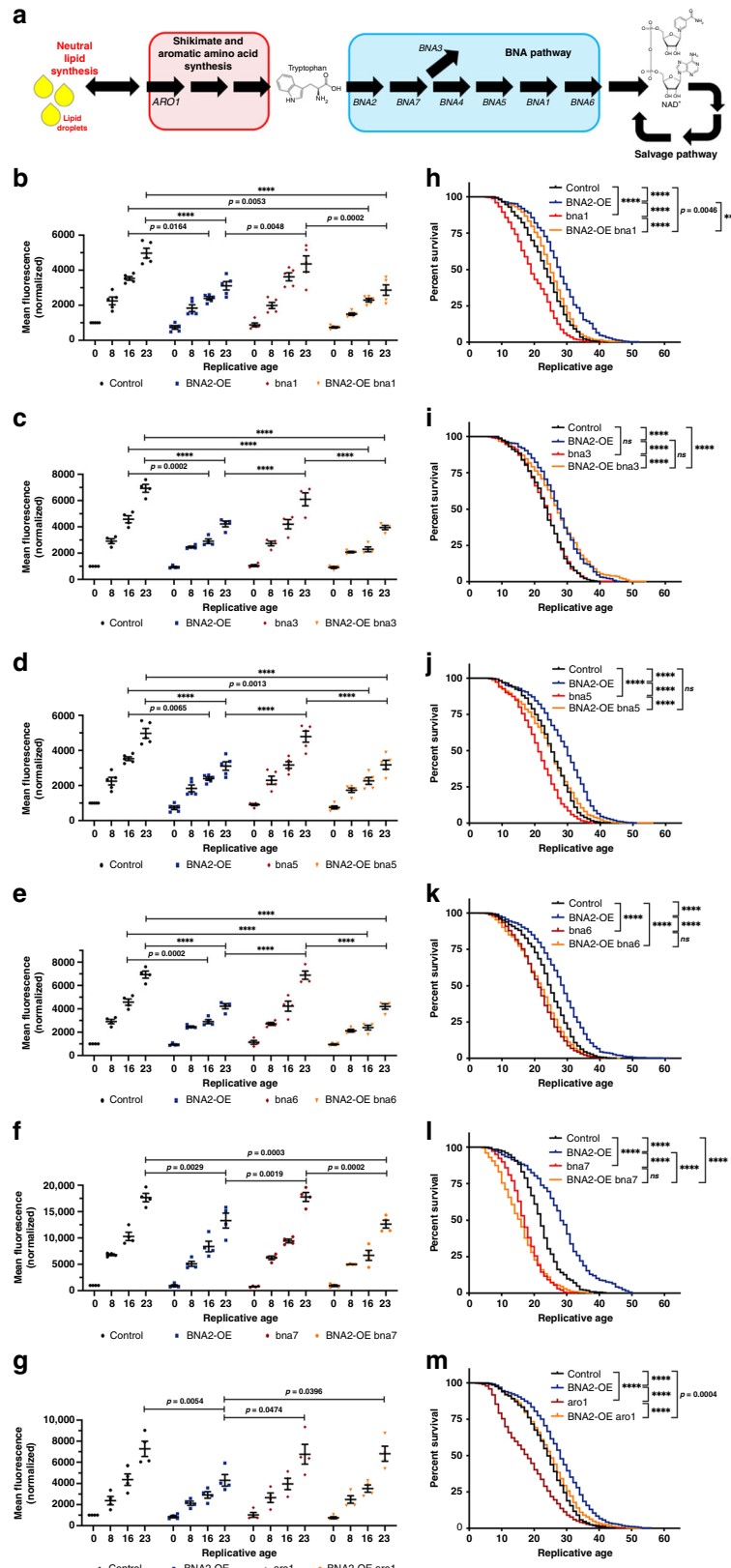

aging have been observed in *Drosophila* and mice[27]. This leaves open the possibility that neutral lipid accumulation provides a selective advantage to aging cells experiencing variable environmental conditions.

We provide evidence that aging cell metabolism can be rewired by *BNA2* overexpression to increase flux through the SA pathway

and thus suppresses LD accumulation during aging. In addition to the pathways reported here, a wider network of genes and reactions are linked to yeast lipid metabolism[28]. It remains to be determined how this network may become skewed toward neutral lipid synthesis and LDs during aging. However, our findings may provide insight: the substrates of the shikimate pathway are

**Fig. 2 Lipid droplet accumulation is separable from longevity. a** Shikimate, aromatic amino acid, BNA (SA-BNA) pathways. **b–g** LD quantification by flow cytometry (±SEM, see Supplementary Fig. 1). Two-way ANOVA multiple comparisons: $p > 0.05$, not significant; $p \leq 0.05$ indicated in figure panels; ****$p \leq 0.0001$. All strains: Control (WT, AB18-07), black dots. BNA2-OE, blue squares. Deletions, red diamonds. BNA2-OE + deletions, orange triangles. n=number of independent experiments. **b** $n = 5$. **c** $n = 4$. **d** $n = 5$. **e** $n = 4$. **f** $n = 4$. **g** $n = 4$. **h–m** RLS analysis (see Fig. 1f, Methods). For each panel $n =$ number of independent experiments. Strain: total number of cells analyzed over $n$; average median RLS ± SEM; average maximal RLS ± SEM. Line colors: Control (WT, AB18-07), black; BNA2-OE, blue; deletions, red; BNA2-OE + deletion, orange. Log-rank tests: $p > 0.05$, not significant); $p \leq 0.05$ indicated in figure panels; ****$p \leq 0.0001$. **h** $n = 3$. Control: 400; 24 ± 1.2; 39.3 ± 0.7. BNA2-OE: 400; 27.5 ± 0.8; 47.7 ± 2.2. bna1Δ: 400; 19 ± 2.3; 38 ± 1.7. BNA2-OE bna1Δ: 400; 25.6 ± 1.8; 42.6 ± 5.2. **i** $n = 3$. Control: 350; 24 ± 1.2; 38 ± 1.2. BNA2-OE: 350; 27.2 ± 0.9; 45 ± 0.6. bna3Δ: 350; 23.7 ± 0.9; 38.7 ± 2.7. BNA2-OE bna3Δ: 350; 27 ± 0.6; 47.3 ± 4.4. **j** $n = 3$. Control: 475; 25.3 ± 0.3; 46.7 ± 4.7. BNA2-OE: 475; 30 ± 0.6; 49 ± 1.2. bna5Δ: 475; 21.7 ± 0.6; 36.7 ± 0.7. BNA2-OE bna5Δ: 475; 25.3 ± 0.3; 47 ± 4.4. **k** $n = 5$. Control: 775; 24.6 ± 0.2; 42.4 ± 0.8. BNA2-OE: 925; 28.6 ± 0.5; 53.2 ± 2.1. bna6Δ: 700; 21.4 ± 0.5; 37.8 ± 1.4. BNA2-OE bna6Δ: 725; 22 ± 0.5; 39 ± 2.1. **l** $n = 3$. Control: 325; 22 ± 1; 38.7 ± 1.7. BNA2-OE: 325; 28.7 ± 0.3; 49.7 ± 0.3. bna7Δ: 325; 17 ± 0.6; 30.7 ± 1.2. BNA2-OE bna7Δ: 325; 15.3 ± 0.9; 33 ± 2.5. **m** $n = 5$. Control: 725; 24.6 ± 0.5; 41.8 ± 1.1. BNA2-OE: 725; 28.6 ± 0.8; 48.6 ± 2.9. aro1Δ: 775; 17.2 ± 2.0; 41.2 ± 2.6. BNA2-OE aro1Δ: 750; 25.0 ± 1.3, 48.2 ± 1.7. Source data for (**b–m**) provided as a Source data file.

supplied by glycolysis and the pentose phosphate pathway. Glycolysis also supplies the pentose phosphate pathway, and importantly, acetyl-CoA (via pyruvate), a precursor of all lipids. Furthermore, altering glucose levels is known to alter the lipid profile of yeast[29–31]. Thus, we propose that as cells age, glycolytic flux is skewed toward neutral lipid and LDs—presumably through pyruvate and acetyl-CoA (Fig. 3j). By contrast, in cells overexpressing BNA2, glycolytic flux is skewed from pyruvate, acetyl-CoA, and neutral lipid toward the SA-BNA pathway, which would reduce the level of LD accumulation during aging (Fig. 3j).

We show here that BNA2-OE increases kynurenic acid, but how exactly this metabolite suppresses neutral lipid accumulation in aging cells remains to be determined. In this light, it is noteworthy that links have been described between kynurenine metabolites (e.g., kynurenic acid) and lipid metabolism in mammalian cells[32–34]. In particular, the regulation of IDO1 (the homolog of BNA2) activity and its expression is under complex regulation in metazoan cells[34–37]. In certain cancers, IDO1 is overexpressed, and is thought to promote cancer progression in part by decreasing exogenous levels of tryptophan, which serves as an immune detection signal[34,38]. As shown here in yeast, increased kynurenine branch point metabolites (e.g., xanthurenate) correlate with replicative lifespan extension (i.e., increased cell growth and proliferation), and may provide further insights about how IDO1 overexpression promotes cell growth and proliferation and, in turn, cancer progression.

## Methods

**Strains.** All yeast strains used are derivatives of S. cerevisiae S288C[39] and UCC8773[19] and listed in Supplemental Table 1. Gene deletion strains were created by one step PCR-mediated gene replacement[39,40]. Plasmid templates for gene deletion construction were from the pRS400 vector[39]. Oligonucleotides for gene replacement and mCherry tagging are listed in Supplementary Table 2. Transformation and insertion of NotI-digested plasmid pAG306-GPD-xsome1 or pAG306-GPD-BNA2-xsome1 was done to create overexpression strains[12]. The parental genotype for all BNA2 overexpression and deletion strains is MATa/MATα his3△1/his3△1 leu2△0/leu2△0 ura3△0/ura3△0 lys2△0/+ ho△::SCW11pr-Cre-EBD78-NATMX/ho△::SCW11pr-Cre-EBD78-NATMX loxP-CDC20-Intron-loxP-HPHMX/loxP-CDC20-Intron-loxP-HPHMX loxP-UBC9-loxP-LEU2/loxP-UBC9-loxP-LEU2 Gene Deletion/Gene Deletion GPD-(BNA2)-URA3-chr1/GPD-(BNA2)-URA3-chr1. Strains expressing in-frame, C-terminal chromosomal fusions of ERG6 with mCherry were made by transformation of parental yeast strains with pKT127-mCherry[12].

**Aging cell lipid-droplet suppressor screen.** Mother enrichment program cells were individually transformed in 96-well format with 280 high-copy 2-µm plasmids from a tiled genomic DNA library as previously described[12]. Each plasmid contained a unique sequence-verified genomic DNA fragment that expressed at least one gene previously determined to affect lipid-droplet morphology or glycerolipid metabolism[14–17] and additional genes that were not previously determined to affect these processes (see Supplementary data). To identify genes that suppressed age-induced accumulation of lipid droplets, all plasmid-expressing strains were individually grown in 2 mL of yeast extract peptone 2% glucose

(YEPD) and kanamycin at 30 °C as done previously[12]. Cells were stained with BODIPY 493/503 (ThermoFisher #D3922) or Nile Red (Sigma #N3013), and imaged by fluorescence microscopy. Plasmid containing strains in which 50% of middle-aged cells exhibited young-cell-like levels of lipid droplets were scored as suppressors of age-induced lipid-droplet accumulation. To confirm potential suppressors (e.g., BNA2), candidate genes were transferred using LR clonase from pDONR221 Gateway entry plasmids from a previously described collection (HIP) into pAG306-GPD-ccdB-xsome1[12,41]. Plasmids were integrated into chromosome 1 of control strains after NotI digestion, and strains were aged, purified, and examined by fluorescence microscopy as above (see Supplementary data 1). The integrated suppressor (BNA2) was confirmed by the ability to maintain young-cell-like levels of lipid droplets in at least 50% of middle-aged cells.

**Enrichment of aging cells from cultures using magnetic beads.** Cells stored in 15% (w/v) glycerol at −70 °C were struck onto YEPD agar and grown at RT < 1 day, and colonies were used to inoculate 2 mL YEPD in 16 × 150 mm culture tubes that were grown 27.5 h at 30 °C, 40 rpm (New Brunswick model TC-7 rotor). Saturated cultures were diluted 1:250,000 in 100 mL YEPD in 250-mL flasks, and grown ~15.5 h at 30 °C, 180 rpm (shaker) to ≤2.5 × 10⁶ cells/mL. Lyophilized NHS-activated supraparamagnetic 50-nm beads (Ocean Nanotech; SN00051) were prepared according to manufacturer's instructions, then suspended to 2.5 µg beads/mL PBS (final). 30 × 10⁶ cells were centrifuged initially at 1000 × g, 5 min, room temperature (RT), then transferred to microcentrifuge tubes using 1 mL PBS + 2% glucose (PBS-G), and centrifuged 1500 × g, 1 min, RT. The supernatant was aspirated and cells were washed again in PBS-G. After a second wash, centrifugation, and aspiration, 600 µL 50% (w/v) PEG-3350, 400 µL PBS-G, and 10 µL bead slurry (in this order) was gently layered onto the cell pellets, which prevents the beads from damaging the cells. Cells were suspended gently by pipetting until homogenous, and incubated 10 min, with rotation, at RT. Labeled cells were centrifuged 1500 × g, 1 min, RT, the supernatant was aspirated, then the cell pellets were rinsed gently with 1 mL 30% (w/v) PEG-3350 (final) in YEPD. Cells were centrifuged as above, supernatants were aspirated, and cells were suspended in 30 °C YEPD. For aging of cells, 1 L cultures of 30 °C YEPD in 4-L flasks were inoculated with 8 × 10⁶ labeled cells, and incubated on a shaker at 30 °C, 95 rpm, whereas the remaining labeled cell suspension was centrifuged as above, fixed in 4% paraformaldehyde in PBS (10 min at RT), and saved in PBS at 4 °C as the starting culture (median age 0 cells). At 11, 23, and 34 h after inoculation, labeled, aged cells were recovered from the aged cell cultures by centrifugation of the cultures (1000 × g, 5 min), then the pelleted cells were resuspended in 5 mL 30 °C YEPD per culture, and the cell suspension was applied to LS MACS columns (Miltenyi Biotec, 130-042-401) on magnets (columns pre-equilibrated on the magnet using 1 column volume of 30 °C YEPD). The columns were allowed to drain by gravity flow, and were washed with three column volumes of 30 °C YEPD. Columns were then taken off the magnets, and the aged cells were eluted and collected in 30 °C YEPD. For microscopy and flow cytometry analyses, 1–2 × 10⁶ of eluted cells were fixed in 4% paraformaldehyde and saved in PBS as above for later analyses, whereas the remaining eluted cell suspension was used to inoculate 1.5 L 30 °C YEPD to grow from 11 to 23 h, or to inoculate 0.5 L 30 °C YEPD to grow from 23 to 34 h. For NAD⁺/NADH and metabolomics analyses, eluted cells were instead processed as described below.

**NAD⁺/NADH analyses.** Young and aged cells enriched as above were washed with ice-cold Yeast Nitrogen Base +2% glucose (final) (centrifugation at 1500 × g, 1 min, 4 °C). NAD⁺/NADH was analyzed using the NAD/NADH quantitation kit according to the manufacturer's instructions (Sigma-Aldrich, MAK037). Briefly, cells were suspended in extraction buffer, subjected to two freeze thaw cycles on dry ice, and stored at −70 °C or immediately cleared by centrifugation (14,000 × g, 5 min, 4 °C). Supernatants applied to Amicon regenerated cellulose 10,000 NMWL centrifugal filters (Millipore #MRCPRT010) were centrifuged (15,000 × g, 1.5 h, at

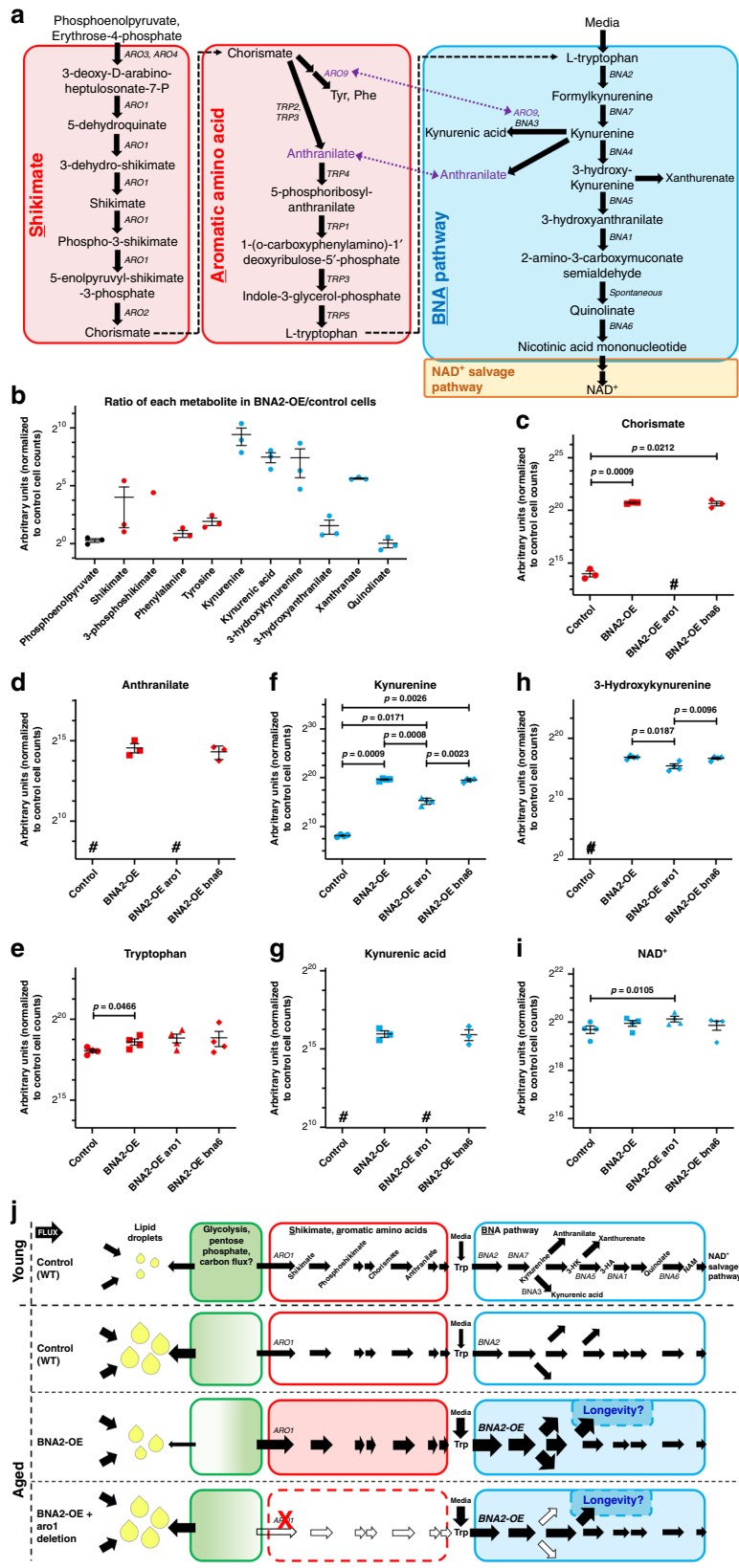

4 °C), and the flow-through was analyzed per manufacturer's instructions [450 nm absorbance measured using a Powerwave XS (Biotek, Winooski, VT)].

**Global and targeted metabolomics**. Aged cells were washed with ice-cold yeast nitrogenous base + 2% glucose (final) as above, then suspended in 80:20 methanol: water at −20 °C, vortexed, and incubated 3 min at −20 °C. Ammonium

bicarbonate pH 6.8 was added to 25 mM final concentration to samples, then incubated at −20 °C for 20 min. Extracts were cleared using Amicon regenerated cellulose centrifugal 10,000 NMWL centrifugal filters (Millipore #MRCPRT010) at 15,000 × $g$, 1.5 h, at −8 °C. Flow-through was kept at −70 °C until analysis. Samples were dried down under nitrogen gas.

Global mass spectrometry analysis using LC-MS was performed at Calico Life Sciences (Fig. 3b, Supplementary Fig. 4, Source data file). Samples were

**Fig. 3 SA pathway metabolite levels anti-correlate with lipid droplet levels during aging. a** The SA-BNA pathway, including metabolite names, *genes*, and sources of crosstalk (purple dashed arrows) between the aromatic amino acid synthesis and BNA pathways. **b–i** Ratios of metabolites from glycolysis, the shikimate and aromatic amino acid pathways, and the BNA pathway are plotted black, red, or blue, respectively. **b** Middle-aged Control (WT, AB18-07) and *BNA2*-OE cells were analyzed by global metabolomics (see Methods). Measurements of specific metabolites (arbitrary units) detected from $n = 3$ independent experiments were normalized to cell counts within each experiment. The normalized measurements from *BNA2*-OE cells were divided by the normalized measurements from Control (WT) cells for each experiment, and these ratios were then averaged and plotted as a ratio of metabolite levels in *BNA2*-OE versus control cells ± SEM (see Supplementary Fig. 4 and source data). **c–i** Cells as indicated were analyzed by targeted metabolomics (see Methods and source data). Measurements of specific metabolites detected in the shikimate (red symbols) or BNA (blue symbols) pathways in Control (WT, AB18-07, dots), *BNA2*-OE (squares), *BNA2*-OE aro1 (triangles), and *BNA2*-OE bna6 (diamonds) cells were normalized to cell counts, averaged from $n = 4$ independent experiments and reported as arbitrary units ± SEM. #Below limit of detection. Paired *t*-test (two-tailed): $p \leq 0.05$ indicated in figure panels; ****$p \leq 0.0001$; $p > 0.05$ indicated as ns (not significant). **j** Model for the mechanism by which *BNA2*-OE suppresses the accumulation of lipid droplets during aging and extends lifespan. Aged Control (WT) cells accumulate lipid droplets during normal aging. *BNA2* overexpression (*BNA2*-OE) decreases lipid-droplet accumulation by pulling some substrates away from lipid droplets through the shikimate and aromatic amino acid (SA) pathways toward the BNA pathway. A deletion in *ARO1* prevents *BNA2*-OE from pulling flux through the SA pathway and from suppressing lipid-droplet accumulation during aging, whereas lifespan extension by *BNA2*-OE correlates with increasing the branch point metabolite xanthurenate. Source data for (**b–i**) are provided as a Source data file.

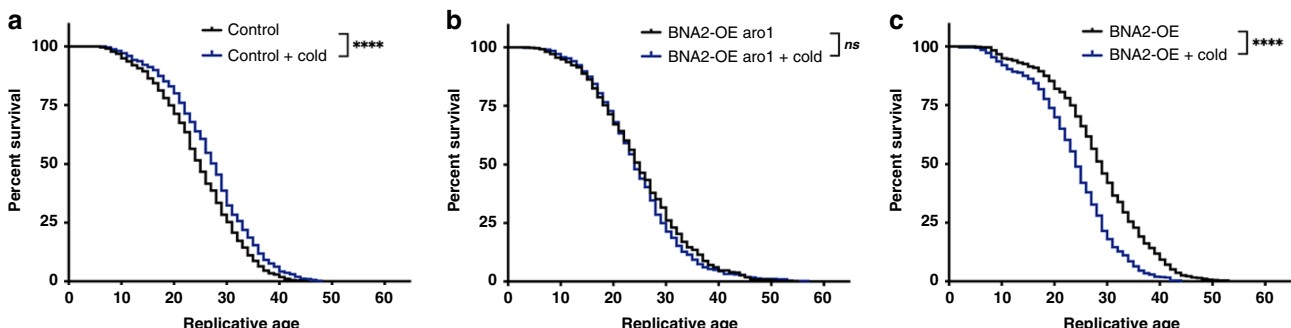

**Fig. 4 Increased survival to cold exposure correlates with LD accumulation during aging. a–c** Cells as indicated were aged in 30 °C media until median age 16 (middle-age), and then exposed to 4 °C (blue lines) or 30 °C (black lines) media at regular intervals until death (see Methods), and lifespan was analyzed as in Fig. 1f (see Methods). Log-rank test: $p > 0.05$, not significant; ****$p \leq 0.0001$. **a** Control (WT, AB18-07, 550 cells at 30 °C, and 525 cells at 4 °C), **b** *BNA2*-OE aro1Δ (525 cells at 30 and 4 °C), and **c** *BNA2*-OE (550 cells at 30 and 4 °C) were analyzed over $n = 4$ independent experiments. Average median and maximal lifespans at 30 °C (±SEM): Control 25.8 ± 1.3, 41 ± 1.9; *BNA2*-OE aro1Δ 25.1 ± 2.2, 44.3 ± 4.2; *BNA2*-OE 29.5 ± 1.7, 48.8 ± 2.5. Average median and maximal lifespans at 4 °C (+SEM): Control 27.5 ± 1.3, 43.3 ± 2.5; *BNA2*-OE aro1Δ 25 ± 1.8, 45 ± 4.6; and *BNA2*-OE 24.8 ± 1.3, 40 ± 3.7. Source data for (**a–c**) are provided as a Source data file.

resuspended in water for analysis on a C18 column in negative ion mode, and diluted 1:4 into acetonitrile for analysis on a HILIC column in positive ion mode. All samples were analyzed using two separate LC-MS methods on Vanquish UPLCs coupled to a Q-Exactive Plus mass spectrometers running Thermo Xcalibur 4.0 (version 4.0.27.10) with Thermo Foundation 3.1 SP1 (version 3.1.83.0) (ThermoFisher Scientific, Waltham, MA).

Metabolites analyzed in positive ionization mode were separated using a SeQuant® ZIC®-pHILIC column, 5 µm particle size, 200 Å, 150 × 2.1 mm. Mobile phase A was 20 mM ammonium carbonate in water (pH 9.2); mobile phase B was acetonitrile. The flow rate was 150 µL/min and the gradient was $t = -6$, 80% B; $t = 0$, 80% B; $t = 2.5$, 73% B; $t = 5$, 65% B, $t = 7.5$, 57% B; $t = 10$, 50% B; $t = 15$, 35% B; $t = 20$; 20% B; $t = 22$, 15% B; $t = 22.5$, 80% B; $t = 24$; 80% B. The mass spectrometer was operated in positive ion mode using data-dependent acquisition (DDA) mode with the following parameters: resolution = 70,000, AGC target = 3.00E + 06, maximum IT (ms) = 100, scan range = 70–1050. The MS² parameters were as follows: resolution = 17,500, AGC target = 1.00E + 05, maximum IT (ms) = 50, loop count = 6, isolation window ($m/z$) = 1, (N)CE = 20, 40, 80; underfill ratio = 1.00%, Apex trigger(s) = 3–10, dynamic exclusion(s) = 25.

Metabolites analyzed in negative ionization mode were separated using a reverse phase ion-pairing chromatographic method with an Agilent Extend C18 RRHD column, 1.8 µm particle size, 80 Å, 2.1 × 150 mm. Mobile phase A was 10 mM tributylamine, 15 mM acetic acid in 97:3 water:methanol pH 4.95; mobile phase B was methanol. The flow rate was 200 µL/min and the gradient was $t = -4$, 0% B; $t = 0$, 0% B; $t = 5$; 20% B; $t = 7.5$, 20% B; $t = 13$, 55% B; $t = 15$, 95% B; $t = 18.5$, 95% B; $t = 19$, 0% B; $t = 22$, 0% B. The mass spectrometer was operated in negative ion mode using data-dependent acquisition (DDA) mode with the following parameters: resolution = 70,000, AGC target = 1.00E + 06, maximum IT (ms) = 100, scan range = 70–1050. The MS² parameters were as follows: resolution = 17,500, AGC target = 1.00E + 05, maximum IT (ms) = 50, loop count = 6, isolation window ($m/z$) = 1, (N)CE = 20, 50, 100; underfill ratio = 1.00%, Apex trigger(s) = 3–12, dynamic exclusion(s) = 20. Metabolites were identified by matching fragmentation spectra and retention times from chemical standards that were previously analyzed on the same instrumentation. Global

metabolomics data RAW files were converted to mzXML files using msconvert from the open source software package ProteoWizard, version 3.0.8789 (http://proteowizard.sourceforge.net/); identity and peak integration were manually verified and quantified using the open source software program Maven, version 8.0.2 (https://github.com/eugenemel/maven)[42]. Quantified measurements were analyzed and plotted using Microsoft Excel v16.34 and Graphpad Prism v8.3.1.

Targeted mass spectrometry analysis was performed at the University of Washington Department of Medicinal Chemistry Mass Spectrometry Center (Fig. 3c–i, Source data file). Samples were reconstituted in 50:50 (v/v) mixture of 100% methanol and 0.4% (v/v) acetic acid in water, and injected into a 1290 Agilent UPLC system coupled to an Agilent 6520 Quadrupole Time of Flight (Q-TOF). Features were extracted and analyzed using Agilent MassHunter Data Acquisition and Processing software, and the measurements above the limit of detection are reported as averages ± SEM from four independent experiments in each figure panel. The measurements (volume in arbitrary units) of each feature were normalized based on the number of cells per sample compared with the number of cells in control (WT) strain, and then analyzed and plotted using Microsoft Excel v16.34 and Graphpad Prism v8.3.1. Standards of each reported metabolite were run as a comparison to detect features in experimental samples.

**Fluorescence microscopy**. All images were captured on a Leica Microsystems (Buffalo Grove, IL) DMI6000 running LAS X (Version 3.3.3.16958) equipped with a Leica DFC365 FX camera, GFP (excitation 470/50, emission 525/50), Texas Red 2 (excitation 560/40, emission 645/76), and QAS (filterset: DFTC excitation 350/50, 490/20, 555/26, 645/30, and emission 455/50, 525/36, 605/52, 705/72) filters, and a HC PL APO ×63 oil objective. Fluorescent images were analyzed using Adobe Photoshop CC 2019 v20.0.8.

**Lipid purification and analysis**. UCC4925 cells were grown on YEPD agar and cultured to saturation and then to log phase (<2.0 × 10⁶ cells/mL) as described above. 20 × 10⁶ cells were washed twice in PBS, and then incubated in 3 mg/mL (w/v) EZ-Link Sulfo-NHS-LC-Biotin (Thermo Scientific, 21335) in PBS (30 min,

RT). Cells were pelleted, supernatant was aspirated, and the labeled cells were suspended in 500 mL YEPD. Cultures were incubated at 30 °C in a shaker (95 rpm). At 10.5 h, cell culture density was determined. In total, $20 \times 10^6$ log cells were harvested (centrifugation $1000 \times g$, 5 min, RT), supernatant was removed by aspiration, and cell pellets were flash-frozen in liquid nitrogen (stored at $-70$ °C) (median age 0 cells). The remaining culture was centrifuged ($1000 \times g$, 5 min, 4 °C), and cell pellets were washed twice with ice-cold PBS, and suspended to $2 \times 10^8$ cells/mL in ice-cold PBS; 1/20th volume of azide-free Multimacs Streptavidin Microbeads (Miltenyi Biotec, 130-092-954) was added, and cells were incubated for 30 min at 4 °C with rotation. Cells were washed by centrifugation and PBS at 4 °C as above. Cells were suspended in ice-cold PBS, and suspensions were applied to LS MACS columns (Miltenyi Biotec, 130-042-401) (pre-equilibrated using 5-mL ice-cold PBS) on magnets. The columns were drained by gravity flow, and washed twice with 1 volume of ice-cold PBS. To elute NHS-biotin-labeled cells (original mother cells), columns were removed from the magnet and cells were eluted with 4-mL ice-cold PBS. Cells were pelleted, supernatant was aspirated, and cells were flash-frozen in liquid nitrogen (stored at $-70$ °C) (median age 8 cells).

Phospholipid (PL) and neutral lipid (NL) purification was performed as described[43]. Total lipids were extracted with 2:1 chloroform:methanol for 1 h at room temperature. Next, 0.2 volumes of 0.9% NaCl were added and the mixture was vortexed and allowed to separate for 2 min. The top layer was removed and the bottom layer was dried down under nitrogen. Dried lipids were resuspended in 1 mL of chloroform. Calibrated phospholipid standard (1,2-Dihepatdecanoyl-sn-Glycero-3-Phosphocholine, Avanti Polar Lipids) and TAG standard (tritridecanoin, Nu-Chek Prep) were added to the total lipid mixture before extraction. PLs and NLs were purified by solid-phase exchange (SPE) chromatography. Extracted lipid was resuspended in chloroform and loaded onto SPE columns (100 mg capacity, Fisher Scientific) pre-equilibrated with 3 mL of chloroform. TAGs were eluted first with 3 ml of chloroform. Glycosphingolipids were eluted next with 5 mL of a 9:1 acetone:methanol mixture, and phospholipids were eluted last with 3 mL of methanol. Purified lipids were dried, resuspended in methanol/2.5% $H_2SO_4$, and incubated for 1 h at 80 °C to create FAMEs. FAMEs were analyzed by gas chromatography/mass spectrometry (GC/MS) (Agilent 5975GC, 6920MS) and data were collected using Agilent MSD Chemstation and analyzed using Agilent ChemStation Integrator. To quantify NL and PL yields, total PL and NL were compared with the internally added standards. Data are presented as NL:PL ratio, which was determined by measuring the sum of all fatty acids found in NL fractions versus the sum of all fatty acids found in PL fractions, and measurements were analyzed and plotted using Microsoft Excel v16.34 and Graphpad Prism v8.3.1.

**Flow cytometry**. Cells were stained with BODIPY 493/503 (ThermoFisher #D3922), Wheat Germ Agglutinin Alexa Fluor 647 Conjugate (ThermoFisher #W32466), and SYTOX Blue Dead Cell Stain (ThermoFisher #S34857), and signals were quantified using a Becton Dickinson (BD) FACSCanto II running BD FACSDiva 6.1.3 software with an excitation wavelength of 488 nm and an emission wavelength of 530 nm (FITC, Bodipy), excitation 633 nm and an emission wavelength of 660 nm (APC, WGA-647), and excitation 405 nm and an emission wavelength of 450 nm (PacificBlue, Sytox Blue). The gating used in analyses was defined to include live cells based on the absence of Sytox Blue Staining, presence of WGA-647 staining in aged samples, and excluded particles that were either too small or too large to be living yeast cells, based on the side scatter (SSC-A) versus forward scatter (FSC-A) plots (see Supplementary Fig. 1). The mean BODIPY fluorescence was determined from 10,000 cells, per time point, per experiment. Data were analyzed with the FlowJo v10 software (Becton, Dickinson & Company, Franklin Lakes, NJ). For each experiment, all BODIPY measurements within an individual experiment were normalized to the average of all BODIPY measurements of median age 0 control cells (see Supplemental Fig. 1 for an example analysis of one experiment), and collected data were normalized and plotted using Microsoft Excel v16.34 and GraphPad Prism v8.3.1 (San Diego, CA).

**Replicative lifespan analyses**. Traditional replicative lifespan analysis has been performed using microdissection as previously described[10]. This method was used to distinguish between the lifespans of control (AB18-07) and BNA2-OE (AB18-11) cells (Fig. 1e).

Lifespans were also determined using a modification to the Mother Enrichment Program (MEP), which can distinguish between lifespans[19] (Fig. 1f). The following modifications were made in order to generate data with high statistical power and collect data on hundreds of cells per strain: Strains were grown on YEPD agar at RT < 24 h, then used to inoculate 2 mL of YEPD, and grown individually and exponentially for 27.5 h at 30 °C on a rotor (40 rpm as above), diluted 1:250,000, and then grown exponentially in 25 mL of YEPD (125-mL flasks) for 15.5 h to a maximum density of $\sim 2 \times 10^6$ cells/mL. $2.0 \times 10^6$ cells were washed with PBS-GN (PBS with 2% glucose and 1 mM nicotinic acid final), and labeled with 5 μg/mL (w/v) DyLight 680 NHS-Ester (Thermo Scientific; #46419) in PBS-GN for 5 min at RT. Labeled cells were washed twice with 30 °C YEPD, and $1.0 \times 10^6$ labeled cells were used to inoculate 4.5 mL 30 °C YEPD, and grown for 2 h on a rotor (40 rpm as above). Fifteen microliters of these 2 h cultures were used to inoculate 5.0 mL YEPD with β-estradiol (1 μM final concentration—to induce the MEP) and ampicillin (50 μg/mL final), and incubated as above. The

media was changed every ~9–12 h (pelleting cultures at $1000 \times g$, 5 min, room temperature before removing the supernatant and adding 5.0 mL 30 °C YEPD (1 μM β-estradiol, 50 μg/mL ampicillin final) for 76 h. Cells were collected by centrifugation ($1000 \times g$, 5 min), fixed in 4% paraformaldehyde, and stained with wheat-germ agglutinin Texas Red (Thermo Fischer #W21405) according to manufacturer's instructions. The DyLight 680-labeled cells (original mother cells) were identified by fluorescent microscopy, and wheat-germ agglutinin Texas Red-labeled bud scars were counted (>125 cells were analyzed per strain and experiment). Each lifespan curve in Figs. 1f, 2h–m, 4a–c, and Supplementary Fig. 2b was obtained by concatenating all the budscar counts for each strain from multiple (≥3) biological replicates/experiments, which only aged and compared the specific strains presented within the respective figure panel. The concatenated replicative lifespans from multiple biological replicates for each strain were then analyzed and plotted using Microsoft Excel v16.34 and Graphpad Prism v8.3.1. Independent budscar counting was accurate to within ±1.

**Statistics and reproducibility**. Statistical analyses including two-way ANOVA (multiple comparisons) for flow cytometry, log-rank tests for replicative lifespan comparisons, and paired t-tests (two-tailed) were performed using GraphPad Prism v8.1.1 (San Diego, CA). The calculated p-values are indicated in the figure panels or legends. All figure panels represent data from indicated biological replicates experiments yielding similar results.

**Reporting summary**. Further information on research design is available in the Nature Research Reporting Summary linked to this article.

## Data availability
The authors declare that the data supporting the findings of this study, including the source data for all four figures and four supplemental figures, are available in the Source data file accompanying this paper and from the corresponding author upon request. The metabolomics datasets are included in the Source data file.

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

## Acknowledgements

We thank members of the Gottschling and Parkhurst labs for helpful discussions; T. Senn and D. Whittington (University of Washington, Department of Medicinal Chemistry Mass Spectrometry Center) for targeted metabolomics analysis; S. Dozono for Flow Cytometry assistance; S-W. Chen for gas chromatography mass spectrometry lipid analysis assistance; C. Kenyon, J. Priess, A. Bedalov, T. Tsukiyama, and A. Ramirez for helpful discussions and comments on the paper. This work was supported by NIH grant AG023779 (to D.E.G./S.M.P.), by Chromosome Metabolism and Cancer Training Grant T32CA009657 (to A.O.B.), and by NCI Cancer Center Support Grant P30 CA015704 (shared resources).

## Author contributions

A.O.B., P.B.G., S.M.P., and D.E.G. contributed to the design of experiments, interpretation of the results, and writing of the paper. A.O.B., P.B.G., and C.L.P. carried out all experiments. C.P.O. developed GC–MS-based lipid analysis. M.A.K. and B.D.B. global metabolomics analysis. A.O.B. wrote the original draft.

## Competing interests

The authors declare no competing interests.
