## [Peer Review File · Nature Communications]

Reviewers' comments:

Reviewer #1 (Remarks to the Author):

The authors made a very interesting observation that there is lipid droplet accumulation in ageing cells (chronologically aged yeast cells) and that this is associated with expression level of a gene in the kynurenine pathway, i.e. BNA2. Through over-expression of this gene accumulation of lipid droplets does not happen, and the cells have increased life-span. This is a very interesting observation, but unfortunately as the paper stands now there is no real mechanistic explanation for the observation. The authors do evaluate the expression of other genes in the pathway, and show that deletion of these genes does not affect the observations. They also measured the intracellular levels of kynurenine and find that this is highly elevated in the BNA2 over-expressed cells. This led to an argument that the coupling between the kynurenine pathway and lipid droplet accumulation is due to an increased flux through the first pathway, and this then drains metabolites otherwise needed for lipid metabolism. This argument is questionable as the flux towards lipid metabolism is orders of magnitude larger than the flux through the kynurenine pathway, even when there is a large accumulation of kynurenine in the cells.

The findings are, however, still very interesting, and I think the authors should just study more about the links between lipid metabolism and kynurenine metabolism, and in particular as yeast is an excellent model organism for doing this. Kynurenine is known to be a regulator of lipid metabolism and is down-regulating phospholipid metabolism, which may be the cause of the decrease LD formation. There are a large number of studies on kynurenine metabolism in human cells as this metabolite is a key regulator of the immune system, and the study is therefore highly relevant and interesting, but the authors do not make any of these connections. A good review for the authors to start would be Sas et al. (2018) from *Molecules* 23:E191 (also discussing interactions of mitochondria metabolism and kynurenine metabolism). If the authors can make a solid link between lipid metabolism and kynurenine metabolism, and then further to ageing then they have a very solid story that would be a much interest to the community. But as the story stands now it is not suitable for *Nature Communication*.

Reviewer #3 (Remarks to the Author):

Authors present interesting data and discussion of age associated changes of lipid metabolism to be associated with longevity. The study as well as the presentation of results is convincingly performed as are the conclusions drawn. However, there are a few shortcomings which should be improved.

Number of replicates is sometimes only 3 or even 2 only (e.g. see figs., 3 and 4), which is not really acceptable for drawing any conclusions.

Discussion should include the fact that not all human cell types regulate tryptophan metabolism in a uniform way. Especially the immune system is prone for distinct considerations, since nitric oxide radical formation differs between human and murine monocytes/macrophages., and nitric oxide is crucial in the regulation of tryptophan catabolism when it inhibits expression and function of the initial enzyme of tryptophan metabolism namely indoleamine 2,3-dioxygenase. So results cannot be generalized easily.

Minor: Denominations of compounds in the paper switches between acids derived from tryptophan conversion and their corresponding anions. These denominations should be uniform as long as not explained by different pH.

Reviewer #4 (Remarks to the Author):

Beas et al. demonstrate that replicative aging of budding yeast coincides with an accumulation of

lipid droplets in mother cells. They show that such replicative age-related build-up of lipid droplets can be suppressed, and replicative lifespan can be prolonged in response to an overexpression of the gene BNA2 known to code an enzyme catalyzing the first step of NAD⁺ synthesis. The authors provide evidence that the overexpression of BNA2 extends yeast replicative lifespan not because it suppresses the observed aging-associated accumulation of lipid droplets and neutral lipid deposition but because it redirects the flow of metabolites from lipid synthesis to tryptophan synthesis. They also establish that an excessive deposition of neutral lipid in lipid droplets is an essential contributor to longevity assurance in replicatively aging yeast exposed to low temperatures.

The manuscript is clearly written and of high technical quality. The claims are convincing, fully supported by the experimental data and appropriately discussed in the context of previous literature. The authors have been fair in their treatment of previous literature. They provided enough methodological details for the experiments to be reproduced by others. The manuscript provides a critical step towards understanding mechanisms linking replicative cellular aging to age-related and environmentally controlled changes in cellular metabolism. Therefore, the manuscript is suitable for Nature Communications, and I recommend accepting it for publication.

RESPONSE TO REVIEWERS

REVIEWER #1

1) *The authors made a very interesting observation that there is lipid droplet accumulation in ageing cells (chronologically aged yeast cells) and that this is associated with expression level of a gene in the kynurenine pathway, i.e. BNA2. Through over-expression of this gene accumulation of lipid droplets does not happen, and the cells have increased life-span. This is a very interesting observation, but unfortunately as the paper stands now there is no real mechanistic explanation for the observation. The authors do evaluate the expression of other genes in the pathway, and show that deletion of these genes does not affect the observations. They also measured the intracellular levels of kynurenine and find that this is highly elevated in the BNA2 over-expressed cells. This led to an argument that the coupling between the kynurenine pathway and lipid droplet accumulation is due to an increased flux through the first pathway, and this then drains metabolites otherwise needed for lipid metabolism. This argument is questionable as the flux towards lipid metabolism is orders of magnitude larger than the flux through the kynurenine pathway, even when there is a large accumulation of kynurenine in the cells.*

Our study concerns replicative aging, which measures the number of times a mother cell divides (i.e., buds off a daughter cell) before senescence and death, not chronological aging (i.e., how long *non-dividing*, starved cells survive). We realized from the reviewer's comments that we had not made this point sufficiently clear. We now define replicative aging in the first sentence of the results section.

The reviewer comments that the accumulation of lipid droplets does not happen in cells over-expressing *BNA2*, which have increased lifespan. However, it was not our intention to imply this. Our data supports that lipid droplet accumulation does happen in *BNA2* over-expressing cells but at a significantly reduced level compared to WT (control) cells (Fig 1d). We have amended the text to make this clear.

We respectfully disagree with the reviewer that we did not provide a mechanistic explanation for the accumulation of lipid droplets during aging, as we believe we have provided a mechanistic (i.e., genetic) explanation for this accumulation. However, we acknowledge that our mechanistic explanation may not have been conveyed clearly, and this led the reviewer to suggest pursuing alternative mechanisms for reduced lipid droplet accumulation in *BNA2* over-expression cells (e.g., kynurenine metabolite signaling). In our revised manuscript we have updated the text, included a more detailed pathway panel (Fig 3a), and importantly, included a mechanistic model (Fig 3j), which more clearly conveys our explanation based on the following information and our novel findings:

- *ARO1* encodes the rate limiting enzyme of the shikimate pathway, upstream of *BNA2*.
- We show that *BNA2-OE* requires *ARO1* to suppress lipid droplet accumulation during aging (Figure 2g).
- We show that *BNA2-OE* requires *ARO1* to increase the levels of metabolites in the Shikimate and Aromatic amino acid pathways (e.g., chorismate, anthranilate), and in the BNA/kynurenine pathway (e.g., kynurenine, kynurenic acid, 3-hydroxykynurenine) (Fig 3d-i).

Taken together, these data strongly support our mechanistic model that *BNA2-OE* reduces LD accumulation during aging by “pulling” enough substrates from LDs through the upstream shikimate pathway towards the BNA pathway (Fig 3j).

The reviewer comments that while we evaluate the expression of other genes in the BNA pathway, we show that deletion of these genes does not affect the observations. However, we do observe differences. In Fig 2, we show that the deletion of genes in the core BNA pathway (including our new findings e.g., deletion of *BNA7*) specifically decreased longevity, while not altering the accumulation of lipid droplets during aging. These results are central to our major conclusion from Fig 2: the accumulation of lipid droplets during aging and longevity are genetically separable and can be modulated independently. In Fig 2 (and as mentioned above), we also found that deletion of the *ARO1* gene in the shikimate pathway upstream of *BNA2* specifically blocked the ability of *BNA2*-OE to reduce the accumulation of lipid droplets during aging, but did not affect lifespan.

We take the reviewer's point concerning flux. We clarified our logic with updated text and models (Fig 3j), and now acknowledge in the discussion (2nd paragraph) that in addition to the SA-BNA pathways reported here, a wider network of genes and reactions are linked to yeast lipid metabolism. It remains to be determined how during aging this network may become skewed towards neutral lipid synthesis and LDs. However, we believe our findings provide insight:

- Glycolysis and the pentose phosphate pathways provide substrates to the shikimate pathway.
- Glycolysis supplies substrates to the pentose phosphate pathway
- Glycolysis provides acetyl-CoA (via pyruvate), a precursor of all lipids.
- It is known that varying glucose levels alters the lipid profile of yeast.

Based on the above, we propose a model: as cells age, glycolytic flux is skewed slightly over time towards neutral lipid and LDs – presumably through pyruvate and acetyl-CoA (Fig 3j). By contrast, in aging cells over-expressing *BNA2*, glycolytic flux is skewed enough from pyruvate, acetyl-CoA, and neutral lipid towards the SA-BNA pathway, which reduces LD accumulation during aging (Fig 3j). In the manuscript, we now highlight that glycolysis and the pentose phosphate pathways supply substrates to the shikimate pathway in Fig 3a as well as in the text, starting at the bottom of page 5 and in the second paragraph of the discussion. We outline our model in Fig 3j and the discussion (2nd paragraph).

2) *The findings are, however, still very interesting, and I think the authors should just study more about the links between lipid metabolism and kynurenine metabolism, and in particular as yeast is an excellent model organism for doing this. Kynurenine is known to be a regulator of lipid metabolism and is down-regulating phospholipid metabolism, which may be the cause of the decrease LD formation. There are a large number of studies on kynurenine metabolism in human cells as this metabolite is a key regulator of the immune system, and the study is therefore highly relevant and interesting, but the authors do not make any of these connections. A good review for the authors to start would be Sas et al. (2018) from Molecules 23:E191 (also discussing interactions of mitochondria metabolism and kynurenine metabolism). If the authors can make a solid link between lipid metabolism and kynurenine metabolism, and then further to ageing then they have a very solid story that would be a much interest to the community. But as the story stands now it is not suitable for Nature Communication.*

We thank the reviewer for pointing out that kynurenine metabolites have been shown to affect lipid metabolism of mammalian cells. In our revised manuscript we acknowledge links between kynurenine metabolites (e.g., kynurenic acid) and lipid metabolism (discussion, last paragraph). In addition, we have substantially updated our text, including more detailed pathway panels (Fig 3a), and importantly including a model panel (Fig 3j), in order to more clearly convey our mechanistic explanation (see above).

REVIEWER #3

Authors present interesting data and discussion of age associated changes of lipid metabolism to be associated with longevity. The study as well as the presentation of results is convincingly performed as are the conclusions drawn. However, there are a few shortcomings which should be improved. Number of replicates is sometimes only 3 or even 2 only (e.g. see figs., 3 and 4), which is not really acceptable for drawing any conclusions. Discussion should include the fact that not all human cell types regulate tryptophan metabolism in a uniform way. Especially the immune system is prone for distinct considerations, since nitric oxide radical formation differs between human and murine monocytes/macrophages, and nitric oxide is crucial in the regulation of tryptophan catabolism when it inhibits expression and function of the initial enzyme of tryptophan metabolism namely indoleamine 2,3-dioxygenase. So results cannot be generalized easily.

We thank the reviewer for their very supportive comments. We now include additional replicates (n=4) for the cold shock experiments (Fig 4a-c) and for the targeted metabolomics (Fig 3c-i). Importantly, we now highlight significant differences in Fig 3c-i, which strongly corroborate our global metabolomics findings.

We thank the reviewer for pointing out the variety and complexity of options for the regulation of tryptophan metabolism in different cell types. We have updated the Discussion to cite relevant literature about this regulation, including the reviewer's suggestions (last paragraph).

Minor: Denominations of compounds in the paper switches between acids derived from tryptophan conversion and their corresponding anions. These denominations should be uniform as long as not explained by different pH.

We thank the reviewer for noting these differences and have now standardized compound naming conventions throughout the manuscript.

REVIEWER #4

Beas et al. demonstrate that replicative aging of budding yeast coincides with an accumulation of lipid droplets in mother cells. They show that such replicative age-related build-up of lipid droplets can be suppressed, and replicative lifespan can be prolonged in response to an overexpression of the gene BNA2 known to code an enzyme catalyzing the first step of NAD+ synthesis. The authors provide evidence that the overexpression of BNA2 extends yeast replicative lifespan not because it suppresses the observed aging-associated accumulation of lipid droplets and neutral lipid deposition but because it redirects the flow of metabolites from lipid synthesis to tryptophan synthesis. They also establish that an excessive deposition of neutral lipid in lipid droplets is an essential contributor to longevity assurance in replicatively aging yeast exposed to low temperatures.

The manuscript is clearly written and of high technical quality. The claims are convincing, fully supported by the experimental data and appropriately discussed in the context of previous literature. The authors have been fair in their treatment of previous literature. They provided enough methodological details for the experiments to be reproduced by others. The manuscript provides a critical step towards understanding mechanisms linking replicative cellular aging to age-related and environmentally controlled changes in cellular metabolism. Therefore, the manuscript is suitable for Nature Communications, and I recommend accepting it for publication.

We thank the reviewer for their very supportive comments.

REVIEWERS' COMMENTS:

Reviewer #1 (Remarks to the Author):

I think the authors have addressed the reviewers comments very well and the revised version is further improved. I have no further comments

Reviewer #4 (Remarks to the Author):

In the revised manuscript of their paper, Beas et al. addressed all requests and concerns of the Reviewers #1 and #3. The manuscript is clearly written and of high technical quality. The claims are convincing, fully supported by the experimental data and appropriately discussed in the context of previous literature. The authors have been fair in their treatment of previous literature. They provided enough methodological details for the experiments to be reproduced by others. The manuscript provides a critical step towards understanding mechanisms linking replicative cellular aging to age-related and environmentally controlled changes in cellular metabolism. Therefore, in its revised form, the manuscript is suitable for Nature Communications, and I recommend accepting it for publication.